# Thermal Poling of Optical Fibers: A Numerical History

**DOI:** 10.3390/mi11020139

**Published:** 2020-01-27

**Authors:** Francesco De Lucia, Pier J. A. Sazio

**Affiliations:** Optoelectronics Research Centre, University of Southampton, Southampton SO17 1BJ, UK; fdl1c13@soton.ac.uk

**Keywords:** non-linear photonics, optical fibers, thermal poling, numerical analysis

## Abstract

This review gives a perspective of the thermal poling technique throughout its chronological evolution, starting in the early 1990s when the first observation of the permanent creation of a second order non-linearity inside a bulk piece of glass was reported. We then discuss a number of significant developments in this field, focusing particular attention on working principles, numerical analysis and theoretical advances in thermal poling of optical fibers, and conclude with the most recent studies and publications by the authors. Our latest works show how in principle, optical fibers of any geometry (conventional step-index, solid core microstructured, etc) and of any length can be poled, thus creating an advanced technological platform for the realization of all-fiber quadratic non-linear photonics.

## 1. Introduction

Since their early implementation in the 1920s [1,2] and subsequent optimization in the 1970s [3] optical fibers have become the most widespread technological platform for telecommunications, mainly due to their relatively low losses and huge bandwidths which greatly exceed the performances of any other system for the transmission of information [4]. For example, it is possible to dope optical fibers with rare earth ions such as erbium to obtain optical amplifiers [5], with ytterbium or neodymium to create fiber lasers, or to embed Bragg grating mirrors and filters into them [6].

Optical fibers are typically exploited as a reliable technology for non-linear photonic devices based on their higher order intrinsic non-linear susceptibility $\chi^{\left(3\right)}$. This, by definition, requires high laser pump intensities and appropriate phase matching conditions to operate efficiently. Third harmonic generation (THG), optical Kerr effect, self-focusing, intensity dependent refractive index, four-wave mixing (FWM) are some of these $\chi^{\left(3\right)}$ -related effects exploited in all-fiber non-linear devices such as, for example, supercontinuum sources [7]. Nevertheless, the absence of intrinsic second order properties in centrosymmetric materials, such as silicate glasses, does not in the first instance allow for their exploitation in creating parametric effects related to this lower order optical non-linearity [8].

However, in 1991 Myers et al. developed a technique, called thermal poling [9], to permanently create effective second-order susceptibility $\chi_{eff}^{\left(2\right)}$ inside glasses. The method consists in the concomitant heating process of a piece of glass and application of a relatively high static electric field through it. When the glass reaches the temperature where some alkali impurity ions (already included inside the glass matrix) have a non-negligible diffusion and drift mobility, the alkali ions start to electromigrate consequently forming a static electric field which is later frozen-in the glass after it is cooled down and the external electric field is removed. The thermal poling technique, at first adopted for bulk glasses, was later used for optical fibers [10] with the main motivation of overcoming some of the issues typical of the classical approach for the realization of non-linear optical devices, based on the interaction between intense light beams and non-linear crystals (such as for example lithium triborate (LBO), beta-barium borate (BBO) or lithium niobate (LiNbO_3_)). These issues can include thermal instabilities of non-linear crystals when illuminated by very high pump powers [11,12], relatively short interaction lengths between light waves involved in the non-linear process, high costs and low damage thresholds of the non-linear crystals and coupling losses due to the presence of air/non-linear crystal interfaces as well as the onerous requirement for continuous optical alignment necessary in free-space optical setup. The appeal represented by the idea of a new technological platform for the realization of efficient and all-fiber non-linear devices produced a significant scientific effort towards the complete exploitation of the thermal poling technique. Since its first appearance, many papers have been published where continuous improvements of the experimental thermal poling technique are presented. In this work we focus our attention on the chronological development of the theoretical models implemented in the last 25 years to explain the glass chemistry and physics behind thermal poling, with the final aim of shedding more light on the mechanisms involved in the creation of the second order non-linearity and ideally understanding how eventually to push the features of the technique beyond its current limits.

## 2. Early Evidence of Second-Order Non-Linearities in Silica Fibers and Thermal Poling

An amorphous dielectric medium can be considered macroscopically isotropic and centrosymmetric and consequently invariant by parity inversion [13]. This means that a glass, as an amorphous medium, lacks any second order non-linear susceptibility $\chi^{\left(2\right)}$ in the electric dipole approximation, because of the parity invariance [8]. For example, silica optical fibers possess a zero $\chi^{\left(2\right)}$ as evidenced by the absence of any quadratic non-linear effect. However, in the 1980s some quadratic nonlinear phenomena were observed in silica optical fibers excited by the radiation generated by high power lasers, for example, the generation of wavelengths corresponding to sum-frequency radiation. The source adopted was a Q-switched and mode-locked neodymium-doped yttrum aluminum garnet (Nd:YAG) laser at 1.064 µm while the sum-frequency light was generated mixing the light of fundamental wavelength and the Stokes wavelengths generated via Raman inelastic scattering [14,15,16]. It is due to the work of Gabriagues et al. the first ever reported observation of second harmonic generation (SHG) in optical fibers [17], while a few years later Osterberg et al. studied the SHG process produced in a silica fiber with laser pulses characterized by a time duration of 100-130 psec and peak power of 70 kW. They observed that, after constantly illuminating a silica fiber, some SH light was collected at its output and the intensity of the light generated grew after a certain time [18,19]. In a later work, it was reported the possibility of reducing this “preparation” time from hours to minutes by illuminating the fiber not only with fundamental wavelength, but also with the SH one [20].

The SHG produced in optical fibers was explained in two different ways. Farries et al. considered that the existence of a non-linear electric quadrupole susceptibility causes the generation of a feeble SH radiation when elevated intensities of the pump light are used [21]. This process produces the formation of color centers (created where fundamental and SH radiation are in phase) in an axially periodic arrangement [22]. Stolen et al., instead, attribute the SHG to a sort of photoinduced phenomenon forming the $\chi^{\left(2\right)}$. Basically, they assume that the origin of the SHG process is the creation of a DC polarization due to the mix of the fundamental and the SH wavelength (already present inside the fiber or even fed from outside). The polarization is characterized by a certain periodicity and is capable of orienting defects and consequently create a phasematched $\chi^{\left(2\right)}$ [20].

Finally, Kashyap created an experimental setup to produce phase-matched electric field-induced second harmonic (EFISH) in single-mode Germania-doped silica fibers [23] by applying a periodic electric field across the core of an optical fiber. The static field created in the fiber’s core generates a periodic $\chi^{\left(2\right)}\propto \chi^{\left(3\right)}E$. It was possible to tune the period of the electric field simply rotating the electrode by an appropriate angle.

As previously discussed, significant permanent effective second order non-linear susceptibility (≈1 pm/V) in centrosymmetric media such as bulk silica glass was demonstrated by Myers et al. [9]. The technique is defined thermal poling and consists in the application of high electric potentials (3–5 kV) through a piece (thickness of 1.6 mm) of fused silica kept at a temperature between 250 and 325 °C for a temporal interval in the range 15–120 min. After the heating phase, the glass is cooled down to room temperature while the voltage is still maintained. The non-linearity is created permanently only in the first few microns of the sample close to the surface where the anodic potential is applied. The $\chi_{33}^{\left(2\right)}$ value for fused silica was found to be 20% of the typical value of the $\chi_{22}^{\left(2\right)}$ measured for LiNbO_3_. A relevant experimental result obtained by Myers et al. is the strict relationship between the value of $\chi^{\left(2\right)}$ obtained and the concentration of the impurities present in the glass the fiber is made of. This observation suggested that the presence of the impurities is of critical importance to make thermal poling an efficient process. 

### 2.1. First Theoretical Explanation of Thermal Poling: Single-Carrier Model

In 1994 Mukherjee et al. presented a first model to explain the thermal poling process dynamics [24]. The model is based on transport of ionic species together with bonds reorientation and states that, by applying an external electric field, the impurity ions already included in the glass matrix create locally static electric fields capable to orient the bonds (related to impurities or Si-O bonds). The induced $\chi^{\left(2\right)}$ is expressed by:(1)$$\chi^{\left(2\right)}\approx \chi^{\left(3\right)}E_{DC}+\frac{np\beta }{5k_{B}T}E_{DC}$$
where $\chi^{\left(3\right)}E_{DC}$ is the term representing the optical rectification process of third order and $E_{DC}$ is the local field due to the non-uniform charge distribution. The other term of Equation (1) represents the electric-field-induced orientation of the molecular second order hyperpolarizability β, with $k_{B}$ the Boltzmann constant, T the absolute temperature of the sample, p the permanent dipole moment associated with the bond, N the number of dipoles involved in the process and a uniaxial molecular system is assumed for the sake of simplicity (the direction of $E_{DC}$ is fixed). Mukherjee et al. introduced for the first time the concept of depletion region formation. The latter consists in the creation of a space-charge zone, situated in proximity of the anodic electrode, which is emptied of impurities. This portion of the glass includes the negatively charged non-bridging oxygen (NBO^−^) centers. By applying high electric fields at high temperature it is possible to move away the ions originally electrostatically linked to them. Applying the Poisson’s equation it is possible to obtain the electric field in the depletion region, which is given by [24]:(2)$$E_{DC}=\frac{qn}{\varepsilon }\left(a-x\right),\;0<x<a,\;a={\left(\frac{2\varepsilon V}{qn}\right)}^{\frac{1}{2}}$$
where a is the depletion width, n is the concentration of ionized impurities, q is the magnitude of the electronic charge, ε the dielectric constant and V the difference of potential externally applied between the two electrodes. This result has been obtained assuming that the depletion layer width on the anodic side is much greater than the corresponding cathode accumulation layer. This model is based on the assumption that there is only one type of carrier involved in the formation of the depletion region, but a later work of Alley et al. [25] highlighted a series of experimental observations which are incompatible with the single-carrier model, including in particular the observation of multiple time scales for the poling, and the dependence of the non-linearity on the sample thermal poling history.

### 2.2. Multiple-Carrier Model for Space-Charge Region Formation

Although the early experimental results seemed to confirm the formation of a negatively charged region underneath the anodic surface of the bulk silica sample [26], as predicted by the single carrier model described in the Section 2.1, other observations indicated that in a thermal poling process of silica there is something more complex than a simple uniformly negatively charged region. In particular, Kazansky et al. found regions of alternating charge below the anode [27] while Myers et al. found that the depth of the non-linearity generated in poled bulk samples was greater for samples poled for 2 h than for 15 min [9]. If the depletion region was a uniformly negatively charged region and the electric field frozen into the glass was expressed by Equation (2), according to the Equation (1), the $\chi^{\left(2\right)}$ induced would be peaked in the region closest to the anode and not at a certain distance as commonly observed in the early poling experiments [28].

After the work of Alley et al. [25], published in 1998, where the important experimental observation of multiple time scales for the formation of the SH signal was reported, in 2005 Kudlinski et al. realized a more exhaustive description of space-charge region formation and induced second order non-linearity in bulk silica glasses [29]. The samples used in their work were disks of fused silica (Infrasil^TM^) characterized by the presence of some types of impurity carriers (typically Na^+^, Li^+^, K^+^, ...) located in the glass matrix with a concentration value of 1 ppm. Disks of different thickness, sandwiched between two Si electrodes, were heated at 250 °C, and poled at 4 kV for different temporal durations. When the impurity carriers become mobile, a high electric field is applied through them, producing their electromigration toward the cathode of the system. As a consequence, a negative space charge is created underneath the anodic surface, due to the fact that negative charges are motionless in the glass matrix (NBO^−^ centers). A huge electric field similar to the dielectric breakdown field is consequently established within the depletion region and a second order susceptibility is then created:(3)$$\chi^{\left(2\right)}=3\chi^{\left(3\right)}E_{DC}$$
where we are assuming a system unidimensional with the electric fields involved all linearly polarized along the same direction for the sake of simplicity. For the first few seconds of the electromigration process the single-carrier model can be still used to describe the time evolution of the depletion region formation [30], while after a certain time, defined optimal time (t_opt_) [31], it is necessary to use a multiple carrier model to describe the temporal evolution of the poling process. If we consider the fast carriers (impurity charges) and the slow carriers (hydrogenated species) and both the migration and the diffusion phenomena, the equation of continuity and the Poisson’s equation can be written as [29]:(4)$$\frac{\partial p_{i}}{\partial t}=-\mu_{i}\frac{\partial \left(p_{i}E\right)}{\partial x}+D_{i}\frac{\partial^{2}p_{i}}{\partial x^{2}}$$
(5)$$\frac{\partial E}{\partial x}=\frac{q}{\varepsilon }\left[\sum_{i}\left(p_{i}-p_{0,i}\right)\right]$$
where $p_{i}$, $p_{0,i}$ and $\mu_{i}$ are respectively the instantaneous concentration (ions/m^3^), the initial (at t = 0) concentration and the mobility (at the temperature where the poling experiment is realized) of the i^th^ species, q is the electron charge, $\varepsilon =3.8\varepsilon_{0}$ is the permittivity of the medium and $D_{i}=k_{B}T\mu_{i}/q$ is the diffusion constant of the i^th^ species, with $k_{B}$ the Boltzmann constant and T the temperature of the medium. The system of Equations (4) and (5) gives the spatial distribution of the electric field in the sample as function of the poling duration. The assumptions related to the voltage applied are that the potential at the anodic surface (x=0) is $V_{app}$, while the potential at the cathodic surface (x=l) is zero. Therefore, the first boundary condition is:(6)$$\int_{0}^{l}Edx=V_{app}$$

While the impurity charges (such as Na^+^) are already present into the sample with the initial uniform concentration $p_{0,Na^{+}}$, the hydrogenated species possess an initial density $p_{0,H^{+}}=0$ and are injected into the glass with an injection rate which depends on the electric field strength at the anodic surface. Therefore, the second boundary condition can be written as:(7)$${\left(\frac{\partial p_{H^{+}}}{\partial t}\right)}_{x=0}=\sigma_{H^{+}}E\left(x=0\right)$$
where $\sigma_{H^{+}}$ is an adjustable parameter used to describe the charge injection into the glass of the hydrogenated species.

In order to describe the dynamical evolution of the space-charge region, we can assume that, as a consequence of the application of the voltage ($V_{app}$) throughout the whole sample of length l, an electric field is created equal to $V_{app}/l$ and, because $\mu_{Na^{+}}\gg \mu_{H^{+}}$, at first a depleted layer close to the anodic surface of the glass is formed due to the Na ions migrating toward the cathode. The induced electric field at the surface increases and screens the external electric field in the part of the sample placed outside the depletion region. When the space charge region is completely created, the maximum value of $E_{DC}\approx 10^{9}V/m$ is obtained. At this time the concentration of the injected carriers per second increases rapidly to the value of $7.5\times 10^{-22}m^{-3}s^{-1}$ (according to the equation that governs the injection into the glass of the hydrogenated species, which affirms that the concentration of those species at the anodic surface is linearly proportional to the value of the electric field at the same surface). At the same time, the drift velocity of the injected hydrogenated species $\nu_{H^{+}}=\mu_{H^{+}}E_{DC}$ reaches the same order of magnitude of to the velocity of the Na ions, which are outside the depletion region, where the external electric field is reduced because it is screened by the formation of the space charge. For poling durations longer than few minutes, these injected ions move deeper and deeper into the glass replacing slowly the Na ions removed previously, consequently neutralizing the NBO^−^ centers (refer to [29] and figures in that paper).

## 3. From Poling of Bulk Glasses to Silica Optical Fibers

The first experiment of thermal poling of a silica fiber was reported by Kazansky et al. in 1994 [10], when a D-shaped fused silica Germania-doped step-index fiber was poled using the setup reported in Figure 3a of ref [10].

This poling configuration was adopted until 1995, when a twin-hole step-index silica fiber was poled for the first time by applying a voltage between the electrodes embedded respectively into the two cladding channels of the fiber [32]. This “twin-hole” fiber became the most adopted geometry for thermal poling of optical fibers [33,34].

After the early works on the poling of silica twin-hole fibers, many other works were published on this topic, such as for example Wong et al. [35], who revealed for the first time the existence, in a poled fiber, of the frozen-in electric field $E_{DC}$, using a Mach–Zehnder interferometer. The technique adopted allowed them to measure both the magnitude and the direction of the frozen-in field. They also measured the third-order non-linearity $\chi^{\left(3\right)}$ of unpoled and poled fibers, concluding that the $\chi^{\left(3\right)}$ has increased by a factor of 2 after the thermal poling process.

A work of Blazkiewicz et al., published in 2001, shows the effects on the dynamics of the poling process of the inclusion of a deposited doped silicate glass ring or of a borosilicate glass ring inside the anode hole of a poled twin-hole fiber [36]. In particular, they observed that in the case of a doped silicate glass ring there is a rapid saturation of the electro-optic coefficient, while the borosilicate glass ring instead acts as a trap layer that retards the evolution of the growth of the electro-optic coefficient. This work demonstrates that by tailoring the structure of the optical fiber to be poled it is possible to modify significantly the characteristics of the poling process and consequently the properties of the poled fiber.

### 3.1. From Conventional Poling to Cathode-Less Poling

The conventional anode–cathode configuration for thermal poling of silica fibers, shown in Figure 2 of [35], generates a space-charge region exclusively in the region surrounding the channel where the anodic electrode is inserted. The space-charge region can be visually observed quite simply by etching the cleaved end of the poled fiber in Hydrofluoric (HF) acid (diluted at 50% in deionized (DI) water) for 1 min and is reported in Figure 5 of [33]. The anode-cathode configuration has the drawback of the tiny distance (≈10–20 μm) between the two channels, which greatly increases the risk of unwanted electric arcing discharge through the glass as a consequence of the application of elevated voltages.

However, in 2009 Margulis et al. showed that it is possible to make a depletion region develop around both the embedded electrodes by connecting them to the same anodic potential [37]. Figure 1 of reference [37] shows the schematics of the cathode-less poling configuration. The first advantage of this new poling configuration is the possibility of reduction of the risk of electrical breakdown through the fiber. Margulis et al. demonstrated also that the $\chi^{\left(2\right)}$ created via the cathode-less method is larger and more stable than the one created via conventional poling.

In 2012, An et al. reported in [38] a study where four different electrodes configurations were adopted to thermally pole a twin-hole optical fiber, including having only one anode wire inserted in one of the two cladding channels, two anode wires embedded inside both the channels, one cathode wire in one channel, and two cathode wires in the channels, in comparison to the conventional one where each one of the two wires embedded in the two channels was respectively connected to the anode and the cathode. The technique of second harmonic microscopy (SHM) was used to visualize the spatial distribution of the second order non-linearity created inside the poled fibers and to measure their magnitude. The results of this work consisted mainly in the observation that both one- and two-anode configurations gave a strong non-linearity compared with the conventional anode-cathode one. At the same time An et al. observed that the two-anode configuration was more reproducible than the one-anode one; for the one cathode-wire and two-cathode-wire configuration, strong non-linearity in a ring shape concentric with the fiber outer surface was induced as if the cathode metal wire were in the center of the twin-hole fiber rather than substantially offset. Figure 1 shows second harmonic (SH) micrographs for the fibers poled in the five different configurations.

In 2014 Camara et al. presented for the first time 2D numerical model of the cathode-less poling technique applied to optical fibers [39]. Their numerical simulations are based on a 2D implementation of the ion-exchange model (the one developed by Kudlinski et al. [29]), applied to poled fibers by using COMSOL^TM^ Multiphysics, and consider the presence in the glass matrix of a faster cation (Na^+^) and a slower cation (Li^+^). Both the ions are assumed to be uniformly distributed in the glass matrix before the poling process starts, while a hydrogenated species (H_3_O^+^) is assumed to be injected from the surfaces in contact with the anodic electrodes. The physics of the 2D model is based on the transport of diluted species and assumes that ions characterized by a low concentration (1 ppm) move in consequence of processes of diffusion and drift due to an electric field [25,29]. The cladding holes of the twin-hole fiber are completely filled by metal [33], providing a perfect equipotential. The equation solved in x, y and t for the concentration of the i^th^ ion (Na^+^, Li^+^ and the hydrogen species, such as H_3_O^+^) is [39]:(8)$$\frac{\partial c_{i}}{\partial t}+\nabla \cdot \left(-D_{i}\nabla c_{i}-z_{i}\mu_{i}Fc_{i}\nabla V\right)=R_{i}$$
where the first term in brackets represents the diffusion while the second term the drift in the electric field E, c is the concentration, D is the diffusivity, *z* is the charge, μ is the ionic mobility, F the Faraday constant, V the electric potential and R the consumption or production rate. The electric field and electric potential distribution are obtained from Maxwell’s equations in the electrostatics regime (magnetic fields are neglected). The boundary conditions assumed in the model of Camara et al. are the initial electrical neutrality of the fiber (the mobile ions and the motionless NBO^−^ centers are characterized by the same concentration inside the fiber), the potential at the surfaces of the holes is the applied voltage during the poling process and zero when the voltage is removed. Furthermore, the external surface of the fiber is at zero volts and that the cations exit it and do not come back. The hydrogenated species is injected from the surface of the cladding holes and move, pushed by the applied electric field. Two possible situations are studied; firstly, where the injection rate of H_3_O^+^ constant, which assumes the presence of ions already at the surface of the hole (in the glass) [25], and secondly where an injection proportional to the electric field on the surface of the hole, as implemented in [29]. The initial carrier concentrations are: c(Na^+^) = 1 ppm uniformly distributed in the glass at t = 0 sec; c(Li^+^) = 1 ppm uniformly distributed in the glass at t = 0 sec; c(H_3_O^+^) = up to 2 ppm injectable from the holes, initially zero inside the entire fiber, with a rate that is either constant, linearly dependent on the field at the electrode edge, or decaying exponentially as the ion supply is exhausted; c(NBO^−^) = 2 ppm uniformly distributed in the glass at t = 0 sec for guaranteeing the initial charge neutrality. It is worth highlighting that the types of charges involved in the poling process, their initial concentration, and their mobility at the desired temperature represent all sources of error in the absolute determination of the precise dynamical evolution of the depletion region. Nevertheless, the results obtained represent a global trend which was strongly validated. Indeed, for the first time in the work of Alley et al. [40] and later in many other papers including the work of Camara et al. [39], the shape of depletion region developed around the anodic electrodes in a thermal poling process is revealed via a process of etching in hydrofluoric acid of the cross section of the fiber. An example of this shape is also reported in Figure 3b in Section 3.2 of this paper. In Figure 2, it is possible to gain an idea of the temporal evolution of the concentrations of the two impurity species already present into the glass matrix (Na^+^ and Li^+^) and of the hydrogenated species injected after the application of the external electric field (H_3_O^+^).

In 2009 another interesting contribution to the understanding of the dynamics of the thermal poling process in silica glass, was given by Zhang et al. [41], who studied multiple poling processes. In particular they demonstrated that the first poling process, in case of a thermal erasure of the non-linearity and subsequent re-poling process, has a strong effect on the latter. Using a two carriers model (the same introduced by Alley et al. [25] and improved by Kudlinski et al. [29]), they quantitatively show that the difference in the evolution of the $\chi^{\left(2\right)}$ is due to the different initial charge distributions before each poling process. The extra hydrogenated species injected during the initial poling process modifies the dynamic of the second poling process; in contrast to the first poling (where the $\chi^{\left(2\right)}$ increases in time), the $\chi^{\left(2\right)}$ tends to decrease in time after reaching a maximum value.

### 3.2. Induction Poling

The cathode-less configuration for poling optical fibers, presented by Margulis et al. in 2009, was adopted until 2014. At that time, De Lucia et al. presented a new technique of thermal poling of silica fibers, called “electrostatic induction” [42,43]. The setup to realize the induction poling process is reported in Figure 3a. Two samples of a twin-hole fused silica fiber, both equipped with solid electrodes embedded in both the cladding channels, are utilized. One of them (≈5 cm of length) is used as electrostatic inductor, while the other one (≈40 cm of length) is the fiber to be poled. The two fibers are kept (on top of a microscope slide placed inside a Petri dish in turn located on top of a heater) adjacent only along the 2.5 cm of the short side of the slide, while the rest of the longer fiber (the one to be poled) is fixed on top of the Petri dish surface with some Kapton tape to facilitate its thermalization. The rear surface of the microscope slide is coated with gold and represents the ground plane of the system. The electrodes embedded in the inductor are both connected to the anodic potential, while the two electrodes embedded in the cladding channels of the fiber to be poled by induction are left floating.

The HF etching, shown in Figure 3b, demonstrated that a depletion region was created all along the whole length of the floating electrodes embedded into the “induced” fiber. The creation of a $\chi_{eff}^{\left(2\right)}$ by induction poling has been also proven measuring a SHG signal produced by pumping the poled fiber with a 1550 nm laser. In order to observe a significant SHG signal it is necessary to periodically erase the non-linearity previously created by thermal poling. The erasure process is obtained by exposing the poled fiber to the light generated by an ultraviolet (UV) source. The periodic erasure allows for obtaining a quasi-phase-matching (QPM) condition between the pump at 1550 nm and the SH light at 775 nm. This induction poling technique thus allows for poling long fibers without any physical contact between the power supply and the embedded electrodes.

Figure 4a,b show the setup used to erase the non-linearity via UV light exposure and then to subsequently characterize the second harmonic signal and the SHG signal peaks obtained for the two different periods of erasure of the non-linearity created in two identical fibers poled under the same experimental conditions. The clear dependence of the wavelength doubled from the period of the erasure confirms that the signal measured is due to a SHG signal created by a quadratic non-linear process via induction poling.

In 2016, De Lucia et al. published the 2D numerical model of the process of creation of a depletion region inside a twin-hole fiber poled via electrostatic induction [44]. The model was inspired by Camara et al. [39], even if it shows some important differences. First of all, the model of Camara et al. assumes that the external surface of the fiber to be poled is always kept at ground potential. While this assumption is suitable for the setup presented by Margulis et al. in [37], the same is not reasonable for a situation where an external field is applied by an inductor to floating electrodes embedded inside the fibers to be poled. If, indeed, the external surface of the fiber to be poled by induction was assumed to be grounded, it would consequently screen the electric field created by the inductor, thus suppressing completely the process of electrostatic induction. Another difference consists in the fact that while in the model of Camara et al. the injection rate of the H_3_O+ ions can be always assumed to be constant, in the thermal poling process via electrostatic induction, the variable floating potentials intrinsic to this process require a field-dependent charge injection. Furthermore, it is necessary to consider the ion recombination process at the cladding–air interface and consequently to modify the field dependency.

The model for the induction poling scheme (whose setup is reported in Figure 3a) is obtained separating the setup in two distinguished parts (indicated in Figure 3a with the letters N and F). In the N part of the setup (the one where inductor and poled fiber are adjacent) the fiber to be poled is immersed in the electric field generated by the inductor, while in the part of the setup where inductor and sample are far from each other (F in Figure 3a) the fiber poled is not immersed in the field lines created by the inductor. This “double” model needs the assumption that the electrodes embedded inside the fiber to be poled are electrically continuous. When the two floating electrodes inserted in the cladding holes of the fiber to be poled are immersed in the external electric field, provided by the inductor in the N region, they become charged as a result of a process of electrostatic induction and reach a specific electric potential. If we assume that there is no drop of electric potential along the floating embedded electrodes, they will be characterized by an equipotential surface along the whole fiber. In other words, whatever is the potential picked up by the floating electrodes in the region N, will be transferred efficiently to any location along the whole fiber.

In the 2D model the two fibers are assumed to be made of two different types of glass. Specifically, the glass the inductor is made of is assumed to be pure silica, which lacks charge impurities, while the fiber to be poled is made of fused silica and is characterized by an initial concentration of 1 ppm of Na^+^ uniformly distributed through its cross section. At the same time, the fiber has up to 1 ppm of H_3_O^+^ ions that can be injected at the cladding holes, while there are no H_3_O^+^ ions inside the fiber at t = 0 s. To initially fulfil the charge neutrality, it is necessary to assume that NBO^−^ centers (characterized by very low mobility) are uniformly distributed inside the fiber with the same concentration of the Na^+^ ions at t = 0 s. The cladding channels of the fiber are considered equipotential. The H_3_O^+^ ions can be injected through the electrode-cladding interface if located at electric potentials higher than the surrounding cladding. A variable parameter $\sigma_{2}$ (whose value is chosen to be identical to that chosen in the model of Kudlinski et al. [29]) is used to describe the charge injection into the sample. The induction poling model considers also the particular case where the electric field is less than zero. In this case, H_3_O^+^ ions close to the cladding (either previously injected or diffused from other regions of the fiber) possess a negative injection rate, which substantially means an outflow. However, if the concentration of H_3_O^+^ is zero at the electrode-cladding interface, the injection rate is zero, even in the case of a “negative” electric field. Therefore, the variation of the injected H_3_O^+^ density per unit of time at the electrode-cladding interface can be expressed by:(9)$${\left(\frac{\partial c_{2}}{\partial t}\right)}_{surface}=\sigma_{2}E,\;E\geq 0\;\mathrm{or}\;E<0\;\mathrm{and}\;c_{2}>0$$
(10)$${\left(\frac{\partial c_{2}}{\partial t}\right)}_{surface}=0,\;\mathrm{otherwise},$$
where $c_{2}$ is the concentration of the H_3_O^+^ species and $\sigma_{2}$ the parameter chosen.

In the near model, the two fibers lie adjacent each other on top of a microscope slide (1 mm thick), with the back face coated with gold and grounded. The far model, on the other hand, consists in the model of Camara [39] (modified according to the considerations reported in the initial part of this section) where the values of the electric potential applied to the two embedded floating electrodes are not constant, but are the values of potential (changing in time) calculated via the near model. Moreover, the far model assumes that the fiber lacks a ground plane. Figure 5 shows the time dynamics of the concentrations of both the “fast” (Na^+^) and “slow” (H_3_O^+^) carriers calculated at three different times of the induction poling process in the near model, while in Figure 6 the concentrations obtained using the far model are reported. It is possible to note that the two depletion regions develop in a different way according to the location where they develop. The reason for this different behaviour is the fact that in the N area the external electric field created by the inductor affects the distribution of the total electric field developed in the region surrounding each electrode, while in the F region the electric field created around each electrode is not modified by the presence of any external electric field. Consequently, even the evolution of the depletion region will be different in each different region of the setup.

### 3.3. Single Anode Poling

The most recent results obtained by De Lucia et al. [45] demonstrated both experimentally and theoretically that the single-anode (S-A) configuration, characterized by the fact that only one electrode is embedded inside one of the two cladding channels of the optical fiber and connected to a certain electrical potential, is superior (in terms of quadratic non-linearity created in the fiber core) to the double-anode (D-A) configuration, introduced for the first time by Margulis et al. [37]. Starting from the theoretical result for a fiber of symmetric geometry (the two cladding channels are at the same distance from the fiber core) and poled in D-A configuration the value of $\chi_{eff}^{\left(2\right)}$ at the center of the fiber is almost zero for long time poling (≈2h). However, it was observed that if only one electrode was connected to the high positive potential while the other electrode was completely removed, the value of the quadratic non-linearity was not null any more at the center of the fiber core. The hypothesis is that the behavior of the D-A configuration is due to the concomitant and competitive evolution of the space-charge formation around the two anodes. In contrast, the S-A poling scheme does not exhibit the same limitation. The two configurations (D-A and S-A) have been then theoretically studied for a fiber of asymmetric geometry (different distance from each cladding hole and the fiber core). Figure 7 shows the trend (simulated by means of COMSOL™ Multiphysics, Edition 5.1, COMSOL, Inc., Burlington, MA, USA) of $\chi_{eff}^{\left(2\right)}$ with the temporal duration of the poling process for both electrode configurations and at five different locations in the fiber core region for an asymmetric geometry of the fiber. The most significant outcome of the numerical simulations consists in the fact that the ultimate value (for long poling times) of $\chi_{eff}^{\left(2\right)}$ in S-A configuration is approximately double if compared to the one calculated in the D-A. The value of the non-linear susceptibility $\chi_{eff}^{\left(2\right)}$ has also been experimentally measured in a process of second harmonic generation (SHG) at 1550 nm in a fiber periodically poled in S-A configuration. The $\chi_{eff}^{\left(2\right)}$ has been periodically erased by exposing the poled fiber to a UV light emitted by a frequency doubled argon-ion laser (CW, 244 nm). The results obtained proved that the S-A scheme for poling silica fibers is preferable to the D-A one in terms of absolute value of $\chi_{eff}^{\left(2\right)}$. Indeed, in their 2019 paper [45], De Lucia et al. also demonstrated that the theoretical result shown in Figure 7 is verified experimentally. Two identical fibers poled in the same experimental conditions and for long time (2 h) but in the two different configurations (D-A and S-A) have been characterized in terms of the $\chi_{eff}^{\left(2\right)}$ obtained in a second harmonic generation process. The value of effective second order non-linear susceptibility obtained for the fiber poled in S-A configuration is double with respect to that obtained for the fiber poled in the D-A configuration. Furthermore, the S-A scheme allows for a significant simplification of the fiber fabrication scheme, as only a single cladding channel will be required for the electrode, thus allowing for with considerably relaxed tolerances on the fiber’s core position relative to the single electrode.

## 4. Conclusions

Thermal poling, a technique invented more than 25 years ago, nowadays still represents an important tool in the area of quadratic nonlinear photonics. More recently, two new developments have been presented. The adoption of liquid electrodes [46] and the demonstration of the induction poling technique [42,44], are potentially useful to create nonlinear quadratic all-fiber devices exploiting different types of waveguides, in terms of length and geometry. In this paper, we have mainly focused our attention on the logical and chronological development of 2D numerical models to explain as deeply as possible the dynamics of evolution of the poling process. In particular, we started from the early theoretical interpretation of the process as based on the electromigration of impurity ions immersed in high electric fields. Later, we presented the step towards a full understanding of the phenomenon, as represented by the work of Kudlinski et al. [29], then further refined by Camara et al. [39], and by the work of De Lucia et al. [44], who applied the 2D model to the induction poling technique, explaining its evolution. Finally, we presented our most recent theoretical work which allowed us to identify in the single-anode configuration the most effective method in terms of absolute value of quadratic non-linearity created inside the glass fiber and also in terms of simplification of the fabrication constraints.

## Figures and Tables

**Figure 1 micromachines-11-00139-f001:**
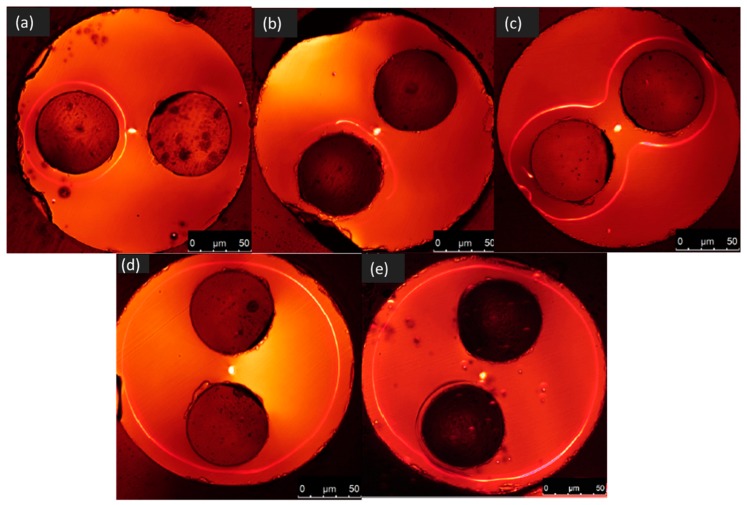
Second harmonic (SH) micrographs of twin-hole silica fibers poled in different electrical configurations, namely (**a**) conventional anode–cathode, (**b**) single anode, (**c**) anode–anode, (**d**) single cathode and (**e**) cathode–cathode. The figures are extracted from the work of An et al. [38].

**Figure 2 micromachines-11-00139-f002:**
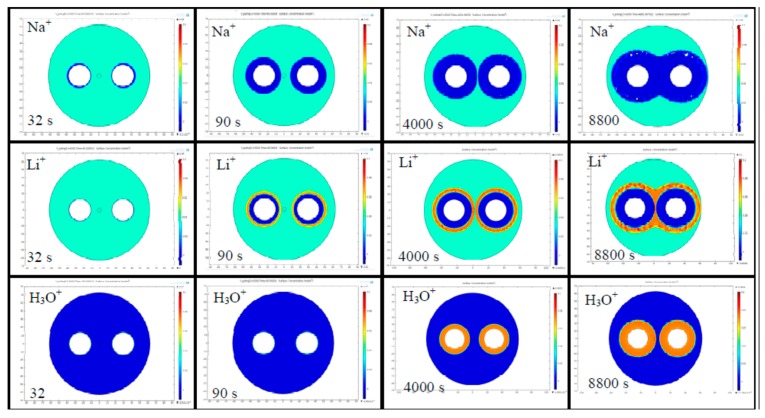
Temporal evolution of the mobile cations for a Germania doped twin-hole fused silica fiber poled in a cathode-less configuration (the two electrodes inserted in the two cladding channels are connected to the same potential of 5 kV). The injection of the H_3_O^+^ ions is considered inexhaustible and capable to neutralize the non-bridging oxygen (NBO^−^) centres depleted of the impurity positive ions moved because of the application of the external electric field [39].

**Figure 3 micromachines-11-00139-f003:**
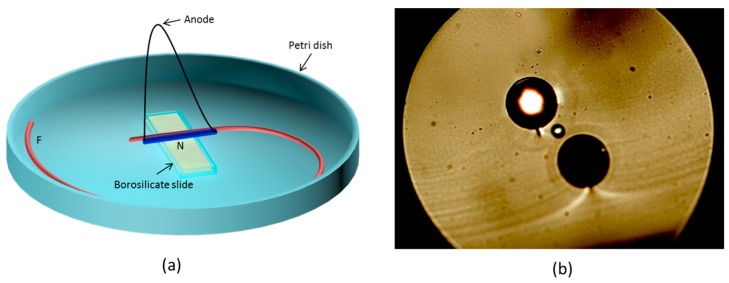
(**a**) Schematic of the setup to pole twin-hole silica fibers via electrostatic induction. The inductor is represented by the blue fiber. Two metallic wires are inserted in the two cladding holes and connected to the identical anodic potential. This fiber is basically used as a layer of dielectric material simply to avoid unwanted electrical arcing discharge in air. The fiber represented in red (whose embedded electrodes are left floating) represents the fiber to be poled. Inductor and sample are attached to a microscope slide by means of some Kapton tape and maintained adjacent along 2.5 cm of the short side of the microscope slide. A gold coating on the backside of the slide (created via e-beam evaporation) works as the ground plane. The Petri dish is maintained at a temperature of ≈ 300 °C during the duration of the process. (**b**) Cross section of a twin-hole silica fiber poled via induction poling technique. The depletion regions are visualized by means of a process of decorative etching in HF acid for 1 min [42,43].

**Figure 4 micromachines-11-00139-f004:**
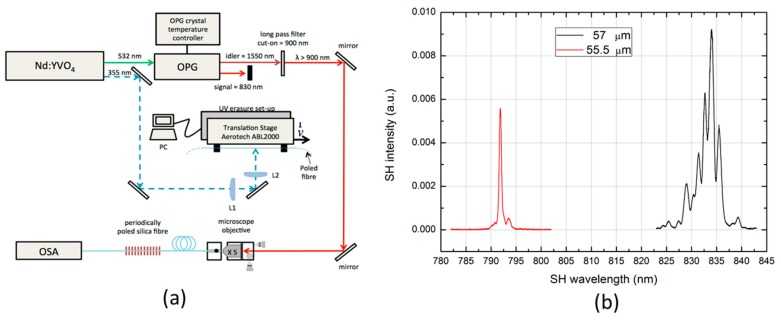
(**a**) Schematic of the setup used to realize the periodic erasure of the quadratic non-linearity created via induction poling. The fiber previously poled via the setup shown in Figure 3a is periodically exposed to ultraviolet (UV) light with the objective of obtaining the quasi-phase matching (QPM) condition between the pump (1550 nm) and the second harmonic generation (SHG) light (775 nm). The wavelength of the laser source used to erase the non-linearity is 355 nm. L1 and L2 stand for the cylindrical lenses of focal lengths f 500 mm and 85 mm, respectively, used to focalize the laser beam in a spot of area of 10 μm × 100 μm at the fiber’s core. Also shown is the setup for the characterization of the SHG signal generated by the periodically poled fiber. (**b**) SHG spectra of induction poled fibers characterized by two different QPM periods of UV erasure [43].

**Figure 5 micromachines-11-00139-f005:**
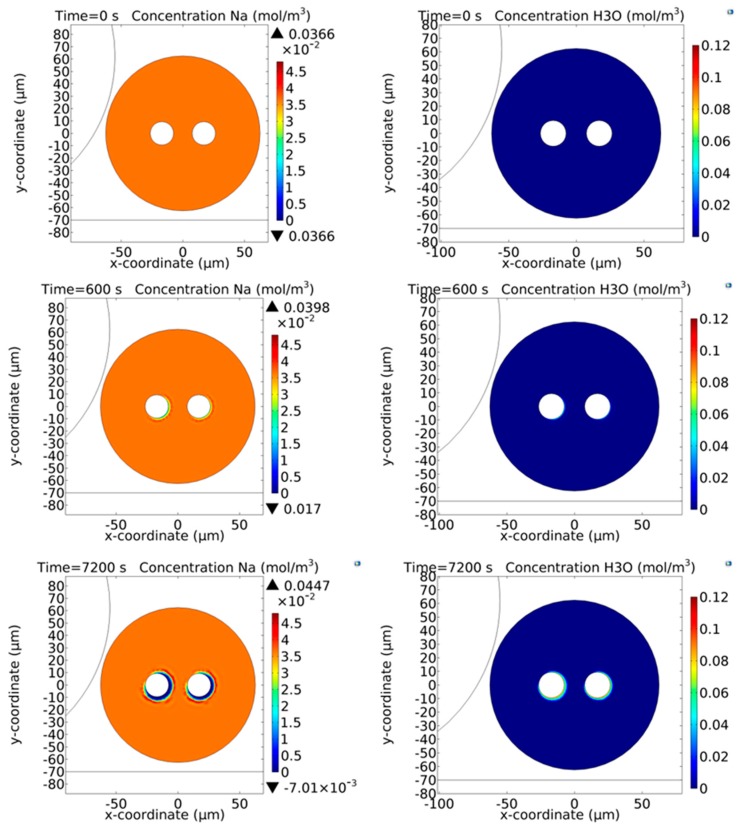
Time dynamics of the Na^+^ ions calculated by means of the near model for induction poling. The Na^+^ ions are considered mobile only in the fiber to be poled, while they are assumed to be motionless inside the inductor. Both the outer surfaces of the two fibers are not assumed to be grounded. The ground of the system is placed at a distance of 1 mm below the microscope slide surface. The concentration of the Na^+^ ions is 1 ppm before the start of the poling process. Both the fibers are considered at a temperature of 300 °C and the injection of the H_3_O^+^ is assumed to be inexhaustible. The H_3_O^+^ ions can consequently neutralize the NBO^−^ centres previously depleted of Na^+^ ions migrated as a result of the application of the external electric field. The two electrodes inserted in the cladding channels of the inductor are both connected to the same electric potential of +5 kV [44].

**Figure 6 micromachines-11-00139-f006:**
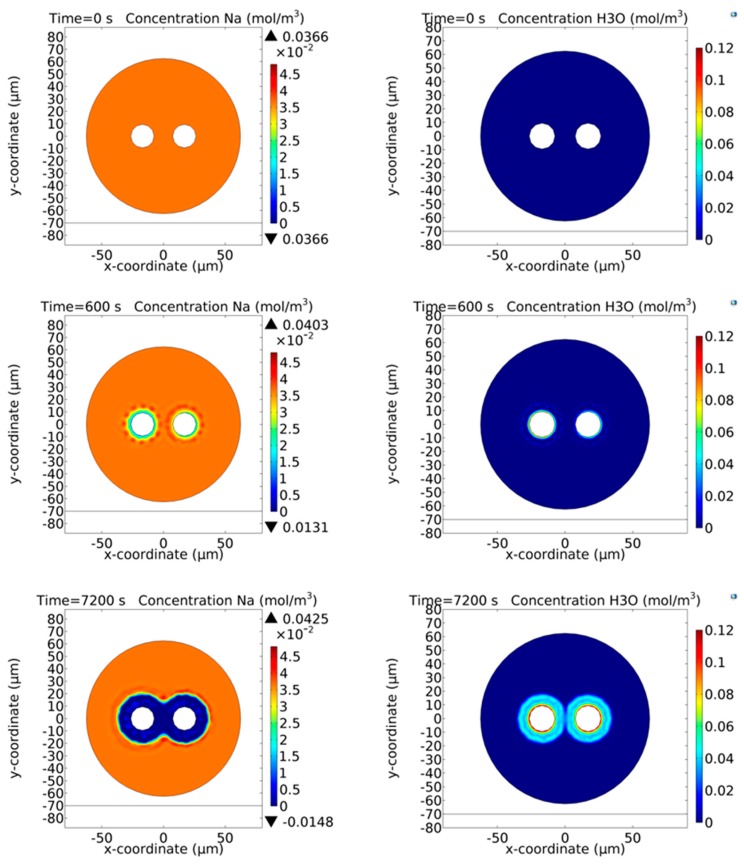
Time dynamics of the Na^+^ ions calculated by means of the far model for induction poling. The two floating electrodes inserted in the cladding channels of the fiber to be poled assume values calculated via the near model in the conditions reported in the caption of Figure 5. The concentration of the Na^+^ ions is 1 ppm before the start of the poling process. The fiber is considered at a temperature of 300 °C and the injection of the H_3_O^+^ is assumed to be inexhaustible [44].

**Figure 7 micromachines-11-00139-f007:**
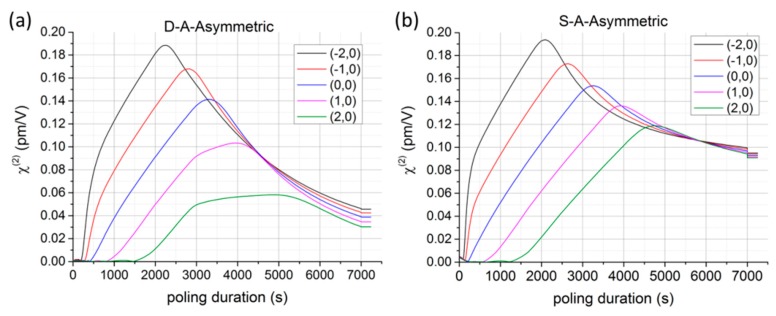
Time evolutions of the $\chi_{eff}^{\left(2\right)}$ numerically obtained for a twin-hole fiber of asymmetric geometry poled in (**a**) D-A and (**b**) S-A configurations. The values of $\chi_{eff}^{\left(2\right)}$ have been calculated via Equation (3) at five different locations in the fiber’s core region (4 μm diameter) and in the plane Y = 0. The legend shows the (x,y) coordinates (in μm) of the points where the values of $\chi_{eff}^{\left(2\right)}$ have been calculated [45].

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
