# Peer review of "Thermal Poling of Optical Fibers: A Numerical History"

_micromachines, 2020, doi:10.3390/mi11020139_

Round 1

Reviewer 1 Report

This manuscript presents a chronological overview of the thermal poling technique, focusing on its application to glass optical fibers. The authors provide some good historical context for the technique along with some light detail on the experimental methods involved, and then devote the bulk of the manuscript to a detailed discussion of physical models that attempt to replicate the experimental observations. Particular emphasis is given in the later paragraphs to their own models, as is their stated intention.

In general, the paper provides a logical and consistent narrative and will be a useful contribution, particularly for researchers entering the field. The language used is mostly good, although the authors should address the issue of overlong paragraphs and several run-on sentences to improve readability. While I appreciate it is some extra effort, I also feel that the addition of two sets of diagrams would really enhance this manuscript. One to detail the various poling techniques under discussion, and the other to describe the various poling mechanisms (models). These figures could then be referenced from the text whenever a new model or technique enters the discussion. I would however not block publication on the absence of these, so I leave it to the discretion of the journal editor whether to require this particular addition.

I have a number of specific suggestions for the authors, which I will break down line-by-line below:

The Introduction section is a single paragraph: I would suggest the authors break it up a little to aid readability. There are a number of run-on sentences which could also be shortened or broken up.

Lines 22-24: Why "moreover"? It would also be useful for the authors to provide a reference to, for example, a review of doped fiber amplifiers, and perhaps one for grating-based fiber devices.

Lines 25-46: The authors make the observation that fibers are not widely used for nonlinear optical devices. While I understand they are focusing on second-order nonlinearity, I think it's only fair at this point to note that nonlinear effects based on third-order susceptibility very commonly exploited in fibers. Perhaps a sentence or two to this effect would be useful in setting the context for the review.

Line 39: "... overcoming some of the issues typical of the classical approach". Perhaps the authors could be a bit more explicit here? I understand an "all-fiber" device has its advantages, but many very capable devices have been built around bulk crystal technologies so the reader may be understandably a little lost by this statement.

Lines 56-58: This is an interesting detail, would the authors consider adding a citation for those unfamiliar with the history?

Line 109: "as described in the section 3.1" should be "section 2.1" I think.

Line 118: "SH signal" -- I don't think second harmonic generation has been mentioned before now, it's common to introduce acronyms prior to their use.

Line 262: the authors state that the results "represent a global trend which is strongly validated". It would be useful to be more precise, and indicate what constitutes "strong validation" -- do they refer to the observations in Figure 1, for example? Or is this just their opinion of the model?

Lines 338-399: This is apparently a single paragraph -- could the authors please take pity on the reader and break it up a little? As it stands it is very hard to read. This paragraph also appears to contain a very high level of detail of the model under discussion: is this really necessary? From the perspective of a reader, a less detailed description might suffice.

Figures 5 and 6: If these are reproduced from reference [28], it should be more explicitly indicated in the figure captions. From a formatting perspective it is preferable if they are arranged so that their captions don't cross page breaks.

Lines 455-456: In the conclusion, the authors claim that the use of liquid electrodes enable the poling of "fibers of any geometry [...] and any length". This is a particularly strong claim that the authors may wish to slightly soften -- one can certainly imagine some complex microstructured geometries which would be quite challenging, and I am not aware that the poling of very long fibers (kilometre-range, for example) has been demonstrated yet.

Reviewer 2 Report

Please see the attached

Round 2

Reviewer 1 Report

The authors have made some significant improvements to the readability of the paper, which are appreciated. The introduction has been improved by the acknowledgement of various chi-3 based nonlinear devices, and I appreciated the additional background provided regarding the discovery of SHG via "self-written" gratings in fibers. The authors also provide a reasonable summary of the challenges faced in creating (chi-2) nonlinear devices using bulk crystals, which I believe readers will appreciate.

While I am a little disappointed the authors chose not to include my suggested figures detailing the poling mechanisms, I stand by my original assertion that the paper would be publishable without them.

Overall this is an improved manuscript, and I am happy to recommend it for publication.

Reviewer 2 Report

The authors have made an attempt to respond to this reviewers comments, however a few questions were glossed over. This however is not so critical as it doesn't affect  the overall quality of the paper, so I recommend its publication pending the editor's final decision.

Coming back to your title, I fail to understand why you find it attractive because of its "potential ambiguity", to put it in your own words. Any ambiguous statement is unclear and confusing, and should be avoided. It's meaning has no "global" connotation.